# Relationships between psychosocial factors during pregnancy and preterm birth in Puerto Rico

Stephanie M. Eick[1,2], John D. Meeker[3], Andrea Swartzendruber[1], Rafael Rios-McConnell[4], Phil Brown[5], Carmen Vélez-Vega[4], Ye Shen[1], Akram N. Alshawabkeh[6], José F. Cordero[1], Kelly K. Ferguson*[3,7]

1 Department of Epidemiology and Biostatistics, College of Public Health, University of Georgia, Athens, GA, United States of America, 2 Program on Reproductive Health and the Environment, Department of Obstetrics, Gynecology, and Reproductive Services, University of California, San Francisco, San Francisco, CA, United States of America, 3 Department of Environmental Health Sciences, University of Michigan School of Public Health, Ann Arbor, MI, United States of America, 4 University of Puerto Rico Graduate School of Public Health, UPR Medical Sciences Campus, San Juan, PR, United States of America, 5 Department of Sociology and Anthropology and Department of Health Sciences, Northeastern University, Boston, MA, United States of America, 6 College of Engineering, Northeastern University, Boston, MA, United States of America, 7 Epidemiology Branch, National Institute of Environmental Health Sciences, Research Triangle Park, NC, United States of America

* kelly.ferguson2@nih.gov

**Data Availability Statement:** All relevant data are within the manuscript and its Supporting Information files.

## Abstract

Psychosocial stress during pregnancy has been associated with adverse pregnancy outcomes including preterm birth (PTB). This has not been studied in Puerto Rico, an area with high PTB rates. Our objective was to develop a conceptual model describing the interrelationships between measures of psychosocial stress and depression, a result of stress, among pregnant women in Puerto Rico and to examine their associations with PTB. We used data from the Puerto Rico Testsite for Exploring Contamination Threats pregnancy cohort (PROTECT, N = 1,047) to examine associations among depression and different continuous measures of psychosocial stress using path analysis. Psychosocial stress during pregnancy was assessed using validated measures of perceived stress, negative life experiences, neighborhood perceptions and social support. Logistic regression was used to examine associations between psychosocial stress measures in tertiles and PTB. Perceived stress, negative life experiences, and neighborhood perceptions influenced depression through multiple pathways. Our model indicated that perceived stress had the strongest direct effect on depression, where one standard deviation (SD) increase in perceived stress was associated with a 57% SD increase in depression. Negative life experiences were directly but also indirectly, through perceived stress, associated with depression. Finally, neighborhood perceptions directly influenced negative life experiences and perceived stress and consequently had an indirect effect on depression. Psychosocial stress was not associated with PTB across any of the measures examined. Our study examined interrelationships between multiple measures of psychosocial stress and depression among a pregnant Puerto Rican population and identified negative neighborhood perceptions as important upstream factors leading to depression. Our findings highlight the

**Funding:** This work was supported by the National Institute of Environmental Health Sciences grants P42ES017198 (PI AA) and P50ES026049 (PI AA) and the National Institutes of Health Office of the Director grants UG3OD023251 (PI AA) and UH3OD023251 (PI AA). Funding was provided by the Intramural Research Program of the NIEHS, NIH (KKF). The funders had no role in study design, data collection and analysis, decision to publish, or preparation of the manuscript.

**Competing interests:** The authors have declared that no competing interests exist.

complex relationship between psychosocial stress measures and indicate that psychosocial stress and depression, assessed using 5 different scales, were not associated with PTB. Future research should investigate other environmental and behavioral risk factors contributing to higher rates of PTB in this population.

## Introduction

Preterm birth (PTB), one of the leading causes of neonatal morbidity and mortality worldwide, disproportionately impacts pregnancies in Puerto Rico. The rates of PTB in Puerto Rico are some of the highest both in the U.S. and globally, with rates as high as 19.9% in 2006[1]. The PTB rate in Puerto Rico declined to 11.9% in 2018, which remains high relative to the mainland U.S. [2]. Some studies have shown that psychosocial stress, as indicated by stressful life events, perceived stress, and depression, is higher among women who go on to deliver preterm [3]. However, associations remain inconsistent and population-specific [3]. In Puerto Rico, pregnant women may also be at a heightened risk for psychosocial stress and clinical outcomes influenced by stress, as an estimated 10% of Puerto Ricans experience major depressive disorder as compared to 8% in the rest of the U.S. [4]. Thus, maternal psychosocial stress during pregnancy may represent a particularly important risk factor for PTB in the Puerto Rican population.

The relationships between measures of psychosocial stress are complex. Psychosocial stress can be triggered from many different sources [5], and is often more prevalent among those with low socioeconomic status (SES) [6]. Increased stressful life events, perceived stress, and a lack of social support are associated with increased symptoms of depression during pregnancy [7–9]. Social support may buffer the effects of stress, although existing studies testing this hypothesis are limited [3]. A growing body of literature also suggests that the quality of one's neighborhood may be a source of psychosocial stress or outcomes influenced by stress. For example, pregnant women living in deprived or lower quality neighborhoods experience stressful life events, inadequate social support, and have high levels of perceived stress, depression, and anxiety [10–13].

The origins of stress and links between different measures of stress during pregnancy have not been explored among Puerto Ricans, but could be important for developing successful interventions to improve pregnancy outcomes.

The purpose of this study was to examine the relationships between different psychosocial stress measurements among pregnant women in the Puerto Rico Testsite for Exploring Contamination Threats (PROTECT) cohort and to investigate the associations of those factors with PTB. Neighborhood perceptions, perceived stress, and negative life experiences were measured to indicate psychosocial stress during pregnancy. We created a conceptual model to test the pathways through which psychosocial stress may influence depression and assessed the role of social support in those relationships. Lastly, we examined associations between these measures and PTB, hypothesizing that increased psychosocial stress and depression would be associated with increased odds of PTB.

## Material and methods

### Study population

Pregnant women included in the present study were enrolled in the PROTECT cohort, an ongoing prospective birth cohort in Northern Puerto Rico that has been previously described in detail [14, 15]. Briefly, we included a subset of women who delivered between January 2011

and September 2017 prior to Hurricane Maria. Women were recruited in early pregnancy prior to 20 weeks gestation from affiliated prenatal clinics. Women were eligible for inclusion in PROTECT if they were between 18–40 years of age, lived in the Northern Karst region, did not use oral contraceptives 3 months prior to conception, did not have *in vitro* fertilization to become pregnant, and were free of known obstetric and medical complications (e.g., diabetes). Women who developed preeclampsia (3%) and gestational hypertension (2.3%) were not excluded from PROTECT. PROTECT was originally designed to examine environmental risk factors for PTB. Therefore, conditions that were *a priori* known to increase the risk of PTB, such as medical conditions and twinning, were excluded to focus on spontaneous PTB, rather than medically-related. Women in PROTECT are invited to complete 3 study visits, occurring between 16–20 weeks gestation, 20–24 weeks gestation, and 24–28 weeks gestation. These study visits were timed to coincide with periods of rapid fetal growth. The Institutional Review Board at all participating locations (University of Michigan, University of Puerto Rico, Northeastern University, and the University of Georgia) approved PROTECT and all women provided written informed consent prior to participation.

**Life Experiences Survey (LES).** Women completed the Life Experiences Survey (LES) at the 2[nd] study visit, which provided information on whether or not they had experienced certain life events (N = 39) anytime in the past year [16]. If they did experience the event, they were asked if it had a negative or positive impact, ranging from extremely negative (a score of -3) to extremely positive (a score of +3). The number of events that each participant perceived as negative (coded -3, -2, -1) were summed to obtain a negative summary measure, indicative of the perceptions of negative events. The absolute value of the summary measure, indicative of perceptions of negative events, was taken to create a positive, continuous measure of negative life experiences (range 0–28). Thus, higher scores were indicative of increased negative life events. Events perceived as positive (a score of +1, +2, or +3) were coded as 0 and had no impact in the current analysis.

**Neighborhood Perceptions (NP).** Also at the 2[nd] study visit, women were asked two questions about perceptions of their neighborhood. Women were first asked if in their opinion, their neighborhood was a very good (a score of 1), good (a score of 2), not very good (a score of 3), or not at all a very good (a score of 4) place to live. Women were then asked if they felt as if their neighborhood was very safe (a score of 1), somewhat safe (a score of 2), somewhat unsafe (a score of 3), or very unsafe (a score of 4). These questions were adapted from the National Children's Study [17]. Responses to both questions were summed to create an overall continuous measure of neighborhood perceptions (NP; range 2–8); thus, higher scores were indicative of negative neighborhood perceptions.

**Perceived Stress Scale (PSS).** The 10-item Perceived Stress Scale (PSS) was administered during the 3[rd] visit. The PSS is designed to measure the extent to which individuals feel that situations in his or her life are stressful [18]. Each item asked about how often specific feelings or thoughts, such as feeling nervous or irritated, occurred within the last month. Responses to each question were ranked on a 5 point Likert scale, with responses ranging from "never" (a score of 0) to "almost always" (a score of 4). Some questions that were positively stated, such as successfully dealing with life hassles, were reverse coded so that higher scores were always associated with increased perceived stress. Responses were summed to create a continuous measure of perceived stress (range 0–40), where higher scores were indicative of increased stress.

**Center for Epidemiologic Studies-Depression (CES-D).** The 20-item Centers for Epidemiologic Studies-Depression (CES-D) scale was also administered at the 3[rd] visit. The CES-D is a screening tool measuring depression symptoms according to the Diagnostic Statistical Manual-IV [19] and has been shown to have good validity among Puerto Rican populations [20]. Questions are designed to measure how often in the past week individuals experience depressive

symptoms. Responses are ranked on a Likert scale and range from "rarely" (a score of 0) to "majority" (a score of 3). Responses were summed to allow for continuous analysis of the depression scale (range 0–48). Higher scores were consistent with increased feelings of depression.

**ENRICHD Social Support Instrument (ESSI).** The Enhancing Recovery in Coronary Heart Disease Patients (ENRICHD) Social Support Instrument (ESSI) was administered during the 3rd visit and is a 7-item scale measuring functional social support [21]. The ESSI is acceptable for use in the general population [22] and has been previously used to assess social support during pregnancy [23]. Women were asked about amount and sources of social support, such as having someone available to listen or provide advice, responses ranged from "none of the time" (a score of 1) to "all the time" (a score of 5). Responses were summed to create a continuous measure of social support (range 8–33), where higher scores were indicative of higher social support.

All psychosocial stress questionnaires were administered in either Spanish or English, depending on participant preference, by trained study staff.

**Gestational age.** Gestational age was assessed using self-reported date of last menstrual period collected at the first study visit and first ultrasound estimates of gestational age per American College of Obstetricians and Gynecologists (ACOG) guidelines [15, 24]. We categorized gestational age into PTB (<37 weeks gestational age) and full term birth (≥37 weeks gestational age).

## Statistical analysis

We examined the means and standard deviations (SD) of the CES-D, ESSI, PSS, LES, and NP across demographic characteristics. For each scale, the overall score was coded as missing if the response to any individual question was missing. Linear regression models were used to determine differences in the CES-D, ESSI, PSS, LES, and NP scales across demographic groups. To examine correlations between each measure, we calculated Pearson's correlation coefficients.

**Path analysis.** Our conceptual path model was developed by reviewing the literature and previously published research. All continuous measures were assessed for normality. Path analysis were used to test our hypotheses using the package 'lavaan' [25] in R Version 3.5.0. Path analysis is an extension of regression analysis which estimates standardized regression coefficients reflecting the direct, indirect, and total effects among variables and evaluates mediation between variables. In path analyses, direct effects indicate the association between two variables where the effect is not mediated through other included variables. Indirect effects show the relationship between one variable and another, through one or more mediating variables. The total effect is the sum of the direct and indirect effects.

The best fitting version of the model was developed through an iterative process where we tested multiple pathways, starting with two variables and gradually adding others. One at a time, we removed those pathways that were non-significant and resulted in poor model fit. Model fit was examined using the chi-square to degree of freedom index ($X^2$/df; values <3 are preferred), Root Mean Square Error of Approximation (RMSEA; values <0.05 are preferred), Standardized Root Mean Square Residual (SRMR; values <0.08 are preferred), Comparative Fit Index (CFI; values >0.9 are preferred), and Tucker-Lewis Index (TLI; values >0.9 are preferred) [26].

When calculating standard errors (SE), we used bias-corrected bootstrapping with 1,000 draws and calculated the corresponding 95% confidence intervals (CI). Missing data in path analyses were analyzed using the full information maximum likelihood (FIML) estimation, which is a recommended way of handling missing data in structural equation modeling [27].

FIML is built into the 'lavaan' package and estimates a likelihood function for all participations based on the non-missing CES-D, ESSI, PSS, LES, and NP measures and covariates for each participant so that all available participants and data are used.

**Moderated mediation.** To test the hypothesis that social support would moderate the associations between psychosocial stress measures and depression, we used the PROCESS macro for SAS 9.4 developed by Hayes. PROCESS is a tool for estimating interactions and the conditional indirect effects of moderated moderation models [28]. Continuous variables were mean centered for moderated-mediation analyses. We calculated regression coefficients for associations between each measure among those who experienced low (one SD below the mean ESSI value; simple slope $a_1$), medium (mean ESSI value), and high (one SD above the mean ESSI value; simple slope $a_2$) social support. PROCESS model 58 allows for multiple mediators and provides 5,000 bootstrapped sample estimates for estimation of indirect effects and 95% bias-corrected bootstrapped CIs. A complete case analysis (N = 841) was used for moderated mediation models.

**Logistic regression.** Logistic regression was used to calculate crude and adjusted odds ratios (OR) and 95% CIs for the associations between individual psychosocial stress measures and depression in tertiles and PTB. Tests for linear trend were conducted using the Cochrane-Armitage test. In logistic regression models, missing data for psychosocial stress measures, depression, and covariates was handled using Multiple Imputation via Chained Equations (mice), in which the independent variables with complete data were used to predict missing values [29]. PTB was not used as a predictor for missing values. We used the package 'mice' in R Version 3.5.0 to produce 10 values for all psychosocial stress measures, depression, and covariates with missing values [29]. Statistical significance was assessed at p-value <0.05.

## Results

There were 1,548 women who were enrolled in the PROTECT cohort prior to September 2017. Of this group, 1,050 had gestational age information available at the time of our analysis. Three women were additionally excluded due to missing information on all covariates. Our final sample size for this analysis included 1,047 women, 107 (10.2%) of which delivered preterm (S1 File Fig A). The highest percentage of women in the PROTECT analytic sample were between ages 18–24 years (38.0%), had received a college degree (43.6%), were employed (62.4%), and were married (56.4%) (Table 1) [15]. Demographics of our analytic sample was similar to those of the overall PROTECT cohort [15] and characteristics of women with complete information on all stress scales is provided in S2 File Table A. Significant correlations were observed between all the CES-D, PSS, LES, NP, and ESSI measures (p-value <0.05 for each correlation) (Table 2). Scores on the PSS, CES-D, LES, and NP were all positively correlated with one another. The strongest correlation observed was between PSS and CES-D (r = 0.65). The ESSI was inversely correlated with each measure, as expected.

Distribution of missingness on the ESSI, PSS, CES-D, LES, and NP scales across demographic characteristics is provided in S2 File Table B. Mean scores on the PSS, LES, and CES-D scales were higher among women who were between ages 18–24, single, currently drinking alcohol, or ever smokers compared to reference groups (S1 File Fig B and Fig C and Fig D). Women with higher stress as measured by NP scale (indicative of increased stress levels) were more likely to be unemployed compared to employed, ever compared to never smokers, and have public compared to private insurance (S1 File Fig E). Women with lower scores on the ESSI (indicative of increased stress) were more likely to be unemployed, single or living with a partner, current or ever smokers, and have public insurance compared to reference groups (S1 File Fig F). Overall, most psychosocial stress variables were associated with lower SES indicators.

**Table 1. Demographic characteristics of the PROTECT study population (N = 1,047).**

| Categorical | N (%) |
|---|---|
| Preterm Birth | |
| Yes | 107 (10.2) |
| No | 940 (89.8) |
| Maternal Age, years | |
| 18–24 | 397 (38.0) |
| 25–29 | 320 (30.6) |
| 30–34 | 214 (20.5) |
| $\geq$35 | 115 (11.0) |
| Maternal Education | |
| <High school | 77 (7.44) |
| High school or equivalent | 132 (12.8) |
| Some college or technical school | 375 (36.2) |
| $\geq$College degree | 451 (43.6) |
| Employment Status | |
| Unemployed | 388 (37.6) |
| Employed | 644 (62.4) |
| Pre-pregnancy BMI | |
| Underweight (<18.5 kg/m$^2$) | 64 (6.46) |
| Normal (18.5-<25 kg/m$^2$) | 492 (49.7) |
| Overweight (25-<30 kg/m$^2$) | 262 (26.5) |
| Obese ($\geq$30 kg/m$^2$) | 172 (17.4) |
| Marital Status | |
| Single | 210 (20.3) |
| Married | 585 (56.4) |
| Living together | 242 (23.3) |
| Alcohol Use | |
| Never | 524 (51.0) |
| Before pregnancy | 442 (43.0) |
| Currently drinking | 62 (6.03) |
| Smoking | |
| Never | 873 (84.2) |
| Ever | 132 (12.7) |
| Current | 32 (3.09) |
| Insurance Status | |
| Public | 364 (35.7) |
| Private | 637 (62.5) |
| Uninsured | 19 (1.86) |
| Continuous | Mean (SD) |
| ENRICHD Social Support Instrument (ESSI) | 27.6 (3.53) |
| Perceived Stress Scale (PSS) | 13.7 (6.84) |
| Center for Epidemiologic Studies-Depression (CES-D) | 11.6 (9.08) |
| Life Experience Survey (LES) | 3.02 (4.03) |
| Neighborhood Perceptions (NP) | 2.53 (0.84) |

Abbreviations: SD, standard deviation; BMI, body mass index.

Note: totals may not sum to 1,047 due to missing values.

**Table 2. Pearson correlation coefficients between psychosocial stress measures.**

|  | CES-D | PSS | LES | ESSI | NP |
|---|---|---|---|---|---|
| Center for Epidemiologic Studies-Depression (CESD) |  | 0.65 | 0.37 | -0.26 | 0.14 |
| Perceived Stress Scale (PSS) |  |  | 0.34 | -0.29 | 0.17 |
| Life Experience Survey (LES) |  |  |  | -0.17 | 0.09 |
| ENRICHD Social Support Instrument (ESSI) |  |  |  |  | -0.16 |
| Neighborhood Perceptions (NP) |  |  |  |  |  |

Note: all correlations are significant at p value<0.05

Marital status, education, and maternal age were *a priori* included as covariates in our path analyses and in logistic regression models based on their known associations with psychosocial stress [12]. CES-D was the primary outcome in our final model and the exposures that demonstrated associations that were greatest in magnitude included the PSS ($\beta = 0.57$, direct path) and the LES ($\beta = 0.18$, indirect path through PSS) (Fig 1 and Table 3). In other words, a one SD increase in perceived stress was directly associated with a 57% SD increase in feelings of depression and a one SD increase in negative life experiences was indirectly associated with a 18% SD increase in feelings of depression. Only the PSS and LES were directly associated with the CES-D.

LES affected the CES-D through both direct ($\beta = 0.15$) and indirect ($\beta = 0.18$) paths, and the indirect effect was greater in magnitude than the direct effect. The LES also had a positive direct effect on the PSS ($\beta = 0.32$).

NP affected the PSS directly ($\beta = 0.12$) and indirectly through LES ($\beta = 0.03$). NP also affected the CES-D indirectly ($\beta = 0.10$) through its effects on PSS and LES scores. Our final model has good fit, as indicated by the model fit statistics all being within the acceptable range. For example, the RMSEA value was 0.00 and the $X^2$/df index was 0.71.

The ESSI was not directly or indirectly associated with the PSS, NP, LES, or CES-D and thus was not included in our final conceptual model. However, in moderated mediation analyses, we found that the relationship between the PSS and CES-D varied based on participants' levels of social support. No moderation by the ESSI was observed for other relationships. To interpret the moderation finding between PSS and CES-D, we plotted estimated levels of CES-D among those with high, medium, and low ESSI scores (S1 File Fig G). Under the condition of low ESSI scores, the indirect effect of NP on CES-D through PSS was greater in magnitude (simple slope $a_1 = 0.85$; 95% CI = 0.76, 0.94) than compared to women with high ESSI scores (simple slope $a_2 = 0.72$; 95% CI = 0.64, 0.82). The full conceptual framework indicating the associations between different parameterizations of stress, confounders, and effect modifiers is shown in S1 File Fig H.

Associations between psychosocial stress, depression, and PTB were null (Table 4). For example, in adjusted analyses women with high compared to low scores on the PSS had no difference in odds of PTB (OR = 0.87; 95% CI = 0.46, 1.63). A 11% increase in odds of PTB was observed among women with high compared to low scores on the LES (95% CI = 0.64, 1.93). High compared to low scores on the CES-D was associated with a 2% decrease in odds of PTB (95% CI = 0.57, 1.69). Tests for linear trend were non-significant across all psychosocial stress measures.

## Discussion

In this study, we examined the relationship between parameterizations of self-reported psychosocial stress and depression in pregnant women from Puerto Rico. We observed that neighborhood perceptions influenced depression through two separate pathways: 1) through increasing negative life experiences and 2) through increasing perceived stress. We also

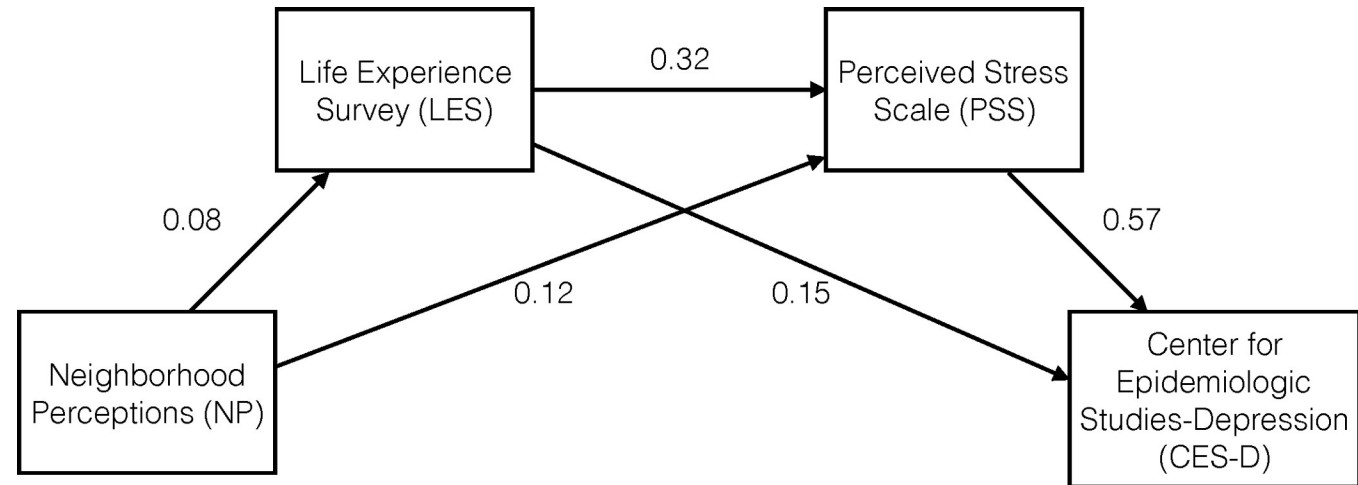

**Fig 1. Path diagram indicating the relationship between psychosocial stress measures and depression in PROTECT study population.** Maternal age, marital status, and education are included as covariates in model (N = 1,047). Note: All paths are significant at p<0.05; missing data handled using full information maximum likelihood. Model fit statistics: $X^2$ = 0.71, p value = 0.40, CFI = 1.00, TLI, 1.02, RMSEA = 0.00, SRMR = 0.00. Abbreviations: $X^2$/df, chi-square to degree of freedom index; RMSEA, Root Mean Square Error of Approximation; CFI, Comparative Fit Index; TLI, Tucker-Lewis Index; SRMR, Standardized Root Mean Square Residual.

showed that in this population perceived stress had the strongest direct effect on depression. None of these measures were associated with PTB.

These findings support a growing body of literature suggesting that the qualities of one's neighborhood may be a source of increased psychosocial stress and depression [10, 11, 30]. Neighborhood perceptions were positively associated with all other metrics of psychosocial stress, including negative life experiences, perceived stress, and ultimately depression. This is in line with previous work showing that women in neighborhoods with high material and social deprivation have increased perceived stress and depression [11]. It is also consistent with a study of African-American women in Michigan which showed that lower levels of perceived neighborhood safety and walkability were associated with increased feelings of perceived stress and depression [30]. Additionally, our findings are supported by a previous study showing that women in disadvantaged neighborhoods experience more stressful life events during pregnancy compared to women in advantaged neighborhoods [10].

**Table 3. Standardized regression coefficients (standard errors) for the best fitting structural equation model of psychosocial stress measures and depression in pregnancy.**

|  | Life Experience Survey (LES) | | |
|---|---|---|---|
|  | **Direct (SE)** | **Indirect (SE)** | **Total (SE)** |
| Neighborhood Perceptions (NP) | 0.08 (0.15) | - | 0.08 (0.15) |
|  | Perceived Stress Scale (PSS) | | |
| Neighborhood Perceptions (NP) | 0.12 (0.25) | 0.03 (0.08) | 0.15 (0.27) |
| Life Experience Survey (LES) | 0.32 (0.06) | - | 0.32 (0.06) |
|  | Center for Epidemiologic Studies-Depression (CES-D) | | |
| Neighborhood Perceptions (NP) | - | 0.10 (0.08) | 0.10 (0.08) |
| Life Experience Survey (LES) | 0.15 (0.07) | 0.18 (0.05) | 0.34 (0.08) |
| Perceived Stress Scale (PSS) | 0.57 (0.04) | - | 0.57 (0.04) |

Note: all paths are significant at p value<0.05; standard errors are estimated using 1,000 bootstrap estimates; missing data handled using full information maximum likelihood specification;—indicates no path; model adjusted for maternal age, marital status, and maternal education.

Abbreviations: SE, standard error.

**Table 4. Crude and adjusted[1] odds ratios of preterm birth (95% confidence intervals) in association with tertiles of psychosocial stress measures and depression in PROTECT (N = 1,047).**

| | Crude | Adjusted[1] |
|---|---|---|
| | OR (95% CI) | OR (95% CI) |
| ENRICHD Social Support Instrument (ESSI) | | |
| High | Ref | Ref |
| Medium | 1.25 (0.72, 2.15) | 1.03 (0.55, 1.95) |
| Low | 1.14 (0.66, 1.97) | 0.87 (0.46, 1.63) |
| p trend | 0.61 | 0.63 |
| Perceived Stress Scale (PSS) | | |
| Low | Ref | Ref |
| Medium | 0.98 (0.57, 1.68) | 0.96 (0.55, 1.67) |
| High | 0.77 (0.45, 1.32) | 0.71 (0.40, 1.23) |
| p trend | 0.35 | 0.23 |
| Life Experience Survey (LES) | | |
| Low | Ref | Ref |
| Medium | 1.29 (0.76, 2.18) | 1.36 (0.79, 2.32) |
| High | 1.06 (0.62, 1.81) | 1.11 (0.64, 1.93) |
| p trend | 0.80 | 0.66 |
| Center for Epidemiologic Studies-Depression (CES-D) | | |
| Low | Ref | Ref |
| Medium | 1.24 (0.75, 2.04) | 1.29 (0.77, 2.14) |
| High | 1.06 (0.63, 1.78) | 0.98 (0.57, 1.69) |
| p trend | 0.82 | 0.96 |
| Neighborhood Perceptions (NP) | | |
| Low | Ref | Ref |
| Medium | 0.93 (0.58, 1.51) | 0.95 (0.58, 1.55) |
| High | 0.74 (0.35, 1.54) | 0.70 (0.33, 1.49) |
| p trend | 0.43 | 0.40 |

Abbreviations: OR, odds ratio; CI, confidence interval; Ref, reference.

[1]Models adjusted for maternal age, education, and marital status.

Note: Psychosocial stress measures were categorized into tertiles indicating high, medium, and low stress. The tertile cut points were as follows: PSS- Low: ≤10, Medium: 11–16, High: >16; CES-D- Low: ≤6, Medium: 7–12, High:>12, LES- Low: 0, Medium: 1–3, High: >3, ESSI- Low: <28, Medium: 28–29, High: >29, NP- Low: ≤2, Medium: 3, High: >3

The direct effect we observed between perceived stress and depression was the greatest in magnitude compared to all other associations in our final model. In addition to a strong direct effect, perceived stress partially mediated the relationships between other psychosocial stress measures (neighborhood perceptions, negative life experiences) and depression. These findings are consistent with another study which demonstrated that perceived stress mediates the relationships between different forms of psychosocial stress and depression [12].

It is hypothesized that psychosocial stress contributes to PTB through activation of the hypothalamic-pituitary-adrenal (HPA) axis, which increases cortisol production [5]. Psychosocial stress may also increase oxidative stress and inflammation, which are increased in mothers who go on to experience PTB [5, 31–33]. Additionally, psychosocial stress may lead to unhealthy behaviors, such as smoking or poor nutrition, which may increase the risk of PTB through these or other pathways [5].

We hypothesized that increased psychosocial stress would be associated with increased odds of PTB, as this has been observed in other studies and is biologically plausible [3, 5]. However, our null associations are consistent with a large body of literature suggesting no association between psychosocial stress or depression and PTB. A recent systematic review examining the association between depression and PTB found that only 25% of studies showed a statistically significant association [34]. Studies of other parameterizations of stress in association with PTB have also been largely null [35–38].

One reason for heterogeneity in these studies may be cross-sectional assessment of stress, as opposed to longitudinal measures from across the life course. In PROTECT, psychosocial stress was measured during pregnancy and focused on self-reported, acute psychosocial stress occurring immediately before (i.e., negative life experiences) and during pregnancy (i.e., neighborhood perceptions, perceived stress, social support). Previous research suggests that women's reproductive health is modified based on early life experiences and cumulative allostatic load, the body's chronic accumulation of stress [6, 39]. Additional research is needed to determine if indices of psychosocial stress from across the life course, such as measures of adverse childhood experiences, are more predictive of birth outcomes as compared to acute stressors included in this study.

Another potential explanation of the inconsistency across studies of stress, depression, and PTB is that these effects may only be observed when stress is sufficiently high. Our study population experienced lower levels of each psychosocial stress indicator and depression relative to other populations. For example, in PROTECT, the mean CES-D score was 11.7 and the mean PSS score was 14.9. Among women enrolled in the Boston Puerto Rican Health Study, the mean CES-D score was 24.4, which is markedly increased compared to those in PROTECT [20]. Similar high scores on the CES-D (mean score of 21.8) were observed among a convenience sample of women recruited from primary care clinics in San Juan, PR [40]. Among women enrolled in the Pregnancy Study Online, an online-based preconception cohort study of women in the U.S. and Canada, the mean PSS score was 15.8, which is also slightly higher than the mean PSS score in PROTECT [41]. The PROTECT cohort is a unique study population and differences in stress levels may also be due to differences across populations or differences in how they respond to questionnaires. Importantly, despite PROTECT women reporting lower levels of psychosocial stress, the relationships observed in our path analysis are consistent with what has been observed in other studies.

Our results should be interpreted in light of some limitations. First, this was an exploratory analysis conducted within an existing study that was designed to address additional research questions and some of our psychosocial stress measures were administered at the same study visit. Our path analysis may be considered cross-sectional and temporality of reporting may be a concern in our study. For example, it is possible that women with depression perceive certain life experiences as more negative [42]. Nonetheless, all associations we identified in our path analysis have been observed in other studies, giving us greater confidence in our results. Second, some women in our study were missing information on psychosocial stress measures, would could have biased our results. However, we used the FIML approach and multiple imputation to address missingness, which should reduce any bias occurring as a result of missing data. Third, women with preexisting medical complications were excluded from our study population, which may have resulted in our study population being inherently healthier than the general population. This may have hindered our ability to detect associations between psychosocial stress and PTB, as previous literature has shown that pregnancy complications are associated with elevated stress levels and women with pregnancy complications are at an elevated risk of PTB [43, 44]. In addition, mediation analysis assumes that there are no unmeasured

confounders, which may not be the case here. Lastly, Spanish translations of the LES, NP, PSS, CES-D, and ESSI have not been validated in Puerto Rican populations.

Despite these limitations, our study has many strengths. The PROTECT cohort is a prospective study and psychosocial stress measures were collected prior delivery. Importantly, we examined several different types of psychosocial stress, each of which are reliable scales used in many other studies. Finally, in the creation of our final conceptual model, we explored several different pathways through which psychosocial stress measures have been associated with one another in the literature, giving us greater confidence in our results.

A unique aspect of this study is that we utilized data on psychosocial stress that was collected prior to Hurricane Maria, but recruitment for the cohort is ongoing. Previous research shows that natural disasters are a source of increased psychosocial stress [45]. It will be important to compare levels of psychosocial stress and the relationships between measures reported here with those collected on PROTECT participants after the Hurricane. Additionally, it will be interesting to examine how the relationship between psychosocial stress and PTB changes before vs. after this event in our study population.

## Conclusions

Our study highlights the complexity of the relationships between different indices of psychosocial stress and depression among pregnant women in Puerto Rico, although none of these measures were associated with PTB. To the best of our knowledge, this is the first study examining psychosocial stress as a risk factor for PTB among Puerto Ricans residing on the island. Stress pathways leading to PTB remain largely misunderstood and our findings may help inform future models that consider diverse sources of psychosocial stress. Future research investigating stress parameterizations in relation to adverse maternal and child health outcomes should explicitly consider the mediating and moderating pathways we identified. Additionally, there may be other pathways leading to elevated stress levels and to adverse birth outcomes, such as anxiety, that were not explored here and warrant exploration. Furthermore, additional work on environmental and behavioral factors is necessary to explain the higher rates of PTB observed in Puerto Rican women.

## Supporting information

**S1 File. Supporting figures for Relationships between psychosocial factors during pregnancy and preterm birth in Puerto Rico.** Fig A. Flow diagram indicating participant selection into final analytic sample.

Fig B. Distribution of Perceived Stress Scale (PSS) across demographic characteristics.

Fig C. Distribution of Life Experience Survey (LES) across demographic characteristics.

Fig D. Distribution of Center for Epidemiologic Studies-Depression (CES-D) across demographic characteristics.

Fig E. Distribution of Neighborhood Perceptions (NP) across demographic characteristics.

Fig F. Distribution of ENRICHD Social Support Instrument (ESSI) across demographic characteristics.

Fig G. Effect of perceived stress on depression moderated by social support.

Fig H. Full model including of all psychosocial stress measures, depression, confounders, and effect modifiers.

(PDF)

**S2 File. Supporting tables for Relationships between psychosocial factors during pregnancy and preterm birth in Puerto Rico.** Table A. Distribution of demographic

characteristics among women with complete information on all psychosocial stress and depression measures (N = 841).

Table B. Distribution of missingness between demographic characteristics and psychosocial stress and depression measures.
(DOCX)

## Author Contributions

**Conceptualization:** Stephanie M. Eick, John D. Meeker, Phil Brown, Akram N. Alshawabkeh, José F. Cordero, Kelly K. Ferguson.

**Data curation:** Phil Brown, Akram N. Alshawabkeh, José F. Cordero.

**Formal analysis:** Stephanie M. Eick, Andrea Swartzendruber, Rafael Rios-McConnell, Ye Shen.

**Funding acquisition:** John D. Meeker, Carmen Vélez-Vega, Akram N. Alshawabkeh, José F. Cordero.

**Investigation:** John D. Meeker, Carmen Vélez-Vega, Akram N. Alshawabkeh, José F. Cordero, Kelly K. Ferguson.

**Methodology:** John D. Meeker, Andrea Swartzendruber, Phil Brown, Kelly K. Ferguson.

**Project administration:** John D. Meeker, Phil Brown, Akram N. Alshawabkeh.

**Resources:** John D. Meeker.

**Software:** José F. Cordero.

**Supervision:** John D. Meeker, Ye Shen, Akram N. Alshawabkeh, José F. Cordero, Kelly K. Ferguson.

**Writing – original draft:** Stephanie M. Eick.

**Writing – review & editing:** Stephanie M. Eick, John D. Meeker, Andrea Swartzendruber, Rafael Rios-McConnell, Phil Brown, Carmen Vélez-Vega, Ye Shen, Akram N. Alshawabkeh, José F. Cordero, Kelly K. Ferguson.

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
