## [Decision Letter · Decision Letter 0]

13 Nov 2019

PONE-D-19-21339

Psychosocial Factors During Pregnancy and Preterm Birth in Puerto Rico

PLOS ONE

Dear Dr. Ferguson,

Thank you for submitting your manuscript to PLOS ONE. After careful consideration, we feel that it has merit but does not fully meet PLOS ONE’s publication criteria as it currently stands. Therefore, we invite you to submit a revised version of the manuscript that addresses the points raised during the review process.

We would appreciate receiving your revised manuscript by Dec 28 2019 11:59PM. To enhance the reproducibility of your results, we recommend that if applicable you deposit your laboratory protocols in protocols.io, where a protocol can be assigned its own identifier (DOI) such that it can be cited independently in the future. For instructions see: http://journals.plos.org/plosone/s/submission-guidelines#loc-laboratory-protocols

We look forward to receiving your revised manuscript.

Kind regards,

Manisha Nair, DPhil, MSc, MBBS

Academic Editor

PLOS ONE

Journal Requirements:

Additional Editor Comments (if provided):

Thank you for submitting this interesting paper. The reviewers have provided detailed comments which I hope will help to improve the paper. I have two comments/ suggestions -

1. Please can the authors show the post-hoc power calculation so that the readers can assess whether the lack of a statistically significant association was due to a small sample size?

2. Please add a Figure of the full conceptual framework showing the pathways of associations between the different parameters as well as confounders and effect modifiers.

Reviewers' comments:

Reviewer's Responses to Questions

**Comments to the Author**

1. Is the manuscript technically sound, and do the data support the conclusions?

Reviewer #1: Yes

Reviewer #2: Yes

2. Has the statistical analysis been performed appropriately and rigorously? 

Reviewer #1: Yes

Reviewer #2: Yes

3. Have the authors made all data underlying the findings in their manuscript fully available?

Reviewer #1: Yes

Reviewer #2: No

4. Is the manuscript presented in an intelligible fashion and written in standard English?

Reviewer #1: Yes

Reviewer #2: Yes

5. Review Comments to the Author

Reviewer #1: It is well established that, different psychosocial stressors have role on preterm delivery. But, cultural variation might be a factor to modify the effect of these stressors. This article, that’s why, wanted to explore the relationships of these stressors with the preterm birth. But, the title of this article is somehow misleading. Because, this title may lead to think that, this article was searching for the names of the stressor only, rather than the relationships.

This study used Puerto Rico Testsite for Exploring Contamination Threats (PROTECT) cohort, which was an ongoing prospective birth cohort in Northern Puerto Rico, to collect samples. Varieties of scales including Life Experiences Survey (LES), Neighborhood Perceptions (NP), Perceived Stress Scale (PSS), Center for Epidemiologic Studies-Depression (CES-D) and ENRICHD Social Support Instrument (ESSI) were used to collect data from the participants. This strengthens the study to found out the relationships with the variables.

The pregnant women from 18-40 years of age lived in the Northern Karst region, with no history of oral contraceptives use 3 months prior to conception, in vitro fertilization, and any known obstetric and medical complications. Data were collected three times during three visits, between 16-20 weeks gestation, 20-24 weeks gestation, and 24-28 weeks gestation. It seems unclear that, on which basis these three gestational ages were chosen.

On 2nd visit they used the LEA and NP, while rests were used in the 3rd visit. They tried to develop a conceptual model at first, to identify the stressors causing depression and assess the role of social support. Then they tried to find out the association between these stressors with PTB. For this, they used the path analysis, where model fit was examined using various statistical measures, including the chi-square to degree of freedom index (X2/df; values <3 are preferred), Root Mean Square Error of Approximation (RMSEA; values <0.05 are preferred), Standardized Root Mean Square Residual (SRMR; values <0.08 are preferred), Comparative Fit Index (CFI; values >0.9 are preferred), and Tucker-Lewis Index (TLI; values >0.9 are preferred). Bias-corrected bootstrapping along with corresponding 95% confidence intervals (CI) were used to define standard errors. The methodology section was written in details, which made the study process more acceptable.

Regarding result, if all of the 1047 women took part in all of the scales then results might be more accurate. Positive correlations were found among the scores on the PSS, CES-D, LES, and NP, where PSS and CES-D had the strongest relation. The ESSI was inversely correlated with each measure, as expected. Findings suggested that, depression had direct association with perceived stress as well as, indirect association with negative life experiences.

Interestingly, this study found no associations between psychosocial stress and PTB. This is actually against to many of the previous evidences. The authors suggested that, this contrast might be due to the low level of stress and having a follow up rather than a cross sectional sample. But, this need to be clarified further or may conduct a separate study addressing this issue.

The authors mentioned some strengths and limitations of this study with adequate explanations. Undoubtedly, this study is well structured, but there are some scopes of improvement. Specially, the writings should be done in a better convenient way. The whole manuscript is difficult to follow because of so many statistical contents. They are necessary, but effort should be given to make it more palatable.

The whole study focused only over the depression as the ultimate pathway to do problem. But, there might be other pathways, like high level of anxiety, those needs to be explored.

Reviewer #2: General comment:

• The way psychosocial stress and depression are discussed is inconsistent throughout the manuscript. In some areas, depression is discussed as a result of stress (one of the research aims of the paper, lines 78-79, 93-94) and in other areas it is considered a measure of stress (lines 71-72, 91-92, 307-308). Similarly, though minor, social support is also occasionally lumped as a ‘psychosocial stress measure’ (e.g., lines 307-308). While this was probably done for ease of communication, it makes it somewhat hard to distinguish whether the references to ‘psychosocial stress’ in the discussion include social support (and also depression).

Methods:

• Line 107 – Were women with preeclampsia/gestational hypertension also excluded? Some studies suggest that stress is associated with preeclampsia/gestational diabetes and development of other medical/pregnancy complications, so this may have hindered your ability to detect an association between stress and PTB and could be included as a limitation. Just speculating, but it may also partially explain why your distribution of CES-D scores was lower than what has been reported in other Puerto Rican samples – your sample is inherently healthier than the general population.

• Line 119-120: The first part of this sentence says that the number of events was summed, but then I’m not sure why you would have to take the absolute value of this – it should already be positive, unless you summed the ‘negativity score.’ Clarification of this sentence would be helpful. You could also include the theoretical range to illustrate how the sum was calculated (0-39 for number of events vs. 0 to 117 for number + severity)

• Line 165-166: Please indicate whether the Spanish translations of survey instruments have been validated.

Results:

• Lines 225-226: It would be helpful to include a study inclusion flowchart.

• Was your sample different than the general PROTECT cohort? You provide descriptive statistics for those with missing data in table S1, but this is only for those who completed ≥1 measure – what about those who didn’t complete any of the five measures?

• Please report the prevalence of PTB in the sample in Table 1. You could also compare this to a recent estimate for Puerto Rico – may help to indicate whether the inclusion criteria resulted in an overall healthier sample than the general population

• Table 4: It would be helpful to include a footnote indicating how the categories were defined for all of the continuous stress measures (tertiles per line 215). It would also be helpful to provide the tertile cutoffs to facilitate comparison to other studies.

Discussion:

• Another limitation is potential selection bias introduced by limiting participants to those without pregnancy/medical complications – excluding those who potentially have more psychosocial stress and are at increased risk of preterm birth.

• May also want to mention strict confounding assumptions for mediation analysis as a limitation

6. PLOS authors have the option to publish the peer review history of their article (what does this mean?). If published, this will include your full peer review and any attached files.

Reviewer #1: Yes: Panchanan Acharjee

Reviewer #2: No

---

## [Author Response · Author response to Decision Letter 0]

20 Dec 2019

Comments from the editor

1. Please can the authors show the post-hoc power calculation so that the readers can assess whether the lack of a statistically significant association was due to a small sample size?

Response: Thank you for considering our manuscript for publication. We have decided to not conduct a post hoc power calculation as there is a consensus in the statistical community that it is incorrect to conduct such power calculations after studies have been conducted. A detailed review of scenarios when it is inappropriate to conduct post hoc power calculations is available at the citation below. 

Hoenig, J. M. and D. M. Heisey (2001). "The Abuse of Power." The American Statistician 55(1): 19-24.

Furthermore this was an exploratory analysis conducted within a study that was designed to address additional research questions We have included the phrase “First, this was an exploratory analysis conducted within an existing study that was designed to address additional research questions” in the discussion section on lines 398-400.

2. Please add a Figure of the full conceptual framework showing the pathways of associations between the different parameters as well as confounders and effect modifiers.

Response: We have added a figure (Figure S8) to illustrate the full conceptual framework. This figure is referenced in the results section on lines 316-318 and includes all stress parameters, confounders, and effect modifiers. 

Reviewer's responses to questions

1. Is the manuscript technically sound, and do the data support the conclusions? The manuscript must describe a technically sound piece of scientific research with data that supports the conclusions. Experiments must have been conducted rigorously, with appropriate controls, replication, and sample sizes. The conclusions must be drawn appropriately based on the data presented.

Reviewer #1: Yes

Reviewer #2: Yes

2. Has the statistical analysis been performed appropriately and rigorously? 

Reviewer #1: Yes

Reviewer #2: Yes

3. Have the authors made all data underlying the findings in their manuscript fully available? The PLOS Data policy requires authors to make all data underlying the findings described in their manuscript fully available without restriction, with rare exception (please refer to the Data Availability Statement in the manuscript PDF file). The data should be provided as part of the manuscript or its supporting information, or deposited to a public repository. For example, in addition to summary statistics, the data points behind means, medians and variance measures should be available. If there are restrictions on publicly sharing data—e.g. participant privacy or use of data from a third party—those must be specified.

Reviewer #1: Yes

Reviewer #2: No

4. Is the manuscript presented in an intelligible fashion and written in standard English?

Reviewer #1: Yes

Reviewer #2: Yes

Comments from reviewers

Reviewer 1

General comment: It is well established that, different psychosocial stressors have role on preterm delivery. But, cultural variation might be a factor to modify the effect of these stressors. This article, that’s why, wanted to explore the relationships of these stressors with the preterm birth. But, the title of this article is somehow misleading. Because, this title may lead to think that, this article was searching for the names of the stressor only, rather than the relationships.

Response: Thank you for your detailed review of our manuscript. We agree that cultural variation is important and that it may be modifying the effect of stress on preterm delivery. We have changed the title of this manuscript to be “Relationships Between Psychosocial Factors During Pregnancy and Preterm Birth in Puerto Rico” in an effort to not mislead readers. 

Specific comments:

1. This study used Puerto Rico Testsite for Exploring Contamination Threats (PROTECT) cohort, which was an ongoing prospective birth cohort in Northern Puerto Rico, to collect samples. Varieties of scales including Life Experiences Survey (LES), Neighborhood Perceptions (NP), Perceived Stress Scale (PSS), Center for Epidemiologic Studies-Depression (CES-D) and ENRICHD Social Support Instrument (ESSI) were used to collect data from the participants. This strengthens the study to found out the relationships with the variables.

Response: We appreciate your thoughtful comments on our manuscript. We agree that including multiple stressors is an important strength of this study. 

2. The pregnant women from 18-40 years of age lived in the Northern Karst region, with no history of oral contraceptives use 3 months prior to conception, in vitro fertilization, and any known obstetric and medical complications. Data were collected three times during three visits, between 16-20 weeks gestation, 20-24 weeks gestation, and 24-28 weeks gestation. It seems unclear that, on which basis these three gestational ages were chosen

Response: The timing of these study visits were chosen to coincide with periods of rapid fetal growth. We have included the sentence “These study visits were timed to coincide with periods of rapid fetal growth.” in the methods section on lines 119-120.

3. On 2nd visit they used the LES and NP, while rests were used in the 3rd visit. They tried to develop a conceptual model at first, to identify the stressors causing depression and assess the role of social support. Then they tried to find out the association between these stressors with PTB. For this, they used the path analysis, where model fit was examined using various statistical measures, including the chi-square to degree of freedom index (X2/df; values <3 are preferred), Root Mean Square Error of Approximation (RMSEA; values <0.05 are preferred), Standardized Root Mean Square Residual (SRMR; values <0.08 are preferred), Comparative Fit Index (CFI; values >0.9 are preferred), and Tucker-Lewis Index (TLI; values >0.9 are preferred). Bias-corrected bootstrapping along with corresponding 95% confidence intervals (CI) were used to define standard errors. The methodology section was written in details, which made the study process more acceptable.

Response: We are glad that you find our statistical methods to be sufficiently detailed. 

4. Regarding result, if all of the 1047 women took part in all of the scales then results might be more accurate. Positive correlations were found among the scores on the PSS, CES-D, LES, and NP, where PSS and CES-D had the strongest relation. The ESSI was inversely correlated with each measure, as expected. Findings suggested that, depression had direct association with perceived stress as well as, indirect association with negative life experiences.

Response: We agree that a limitation of our study was that information on psychosocial stress was missing on some participants. However, given that we used the Full Information Maximum Likelihood approach in the path analysis and used multiple imputation in all other analyses to handle the missingness, we do not believe that our results are substantially biased. Nonetheless, we have noted the missing data on some women as a limitation on lines 405-408 with the below text.

“Second, some women in our study were missing information on psychosocial stress measures, would could have biased our results. However, we used the FIML approach and multiple imputation to address missingness, which should reduce any bias occurring as a result of missing data.”

5. Interestingly, this study found no associations between psychosocial stress and PTB. This is actually against to many of the previous evidences. The authors suggested that, this contrast might be due to the low level of stress and having a follow up rather than a cross sectional sample. But, this need to be clarified further or may conduct a separate study addressing this issue.

Response: We agree that our findings that psychosocial stress was not associated with PTB was in contrast to previous studies. We have clarified that future work is needed to determine if assessment of stress from across the life course, as opposed to a cross-sectional measure of stress, is more strongly associated with PTB. The sentences “Additional research is needed to determine if indices of psychosocial stress from across the life course, such as measures of adverse childhood experiences, are more predictive of birth outcomes as compared to acute stressors included in this study.” are included in the discussion section on lines 379-382.

Additionally, this is a unique study population and it is possible that differences in stress levels are due to differences across populations or differences in how they respond to questionnaires. For example, PROTECT participants have good ongoing relationships with study researchers, so it is possible that the stress of the interviews is lessened. We have noted this with the sentence “The PROTECT cohort is a unique study population and differences in stress levels may also be due to differences across populations or differences in how they respond to questionnaires.” in the discussion section on lines 393-395.

6. The authors mentioned some strengths and limitations of this study with adequate explanations. Undoubtedly, this study is well structured, but there are some scopes of improvement. Specially, the writings should be done in a better convenient way. The whole manuscript is difficult to follow because of so many statistical contents. They are necessary, but effort should be given to make it more palatable.

Response: We have edited our methods and results sections in an effort to make the statistical content easier to follow and understand. We have also added headings within our statistical analyses section.

7. The whole study focused only over the depression as the ultimate pathway to do problem. But, there might be other pathways, like high level of anxiety, those needs to be explored.

Response: We agree that anxiety may be an important pathway that was not explored in our study. However, a measure of anxiety was not available in this study population. We have included the below sentence in the conclusions section on lines 438- 440 to indicate that future research should focus on characterizing additional pathways, such as anxiety. 

“Additionally, there may be other pathways leading to elevated stress levels and to adverse birth outcomes, such as anxiety, that were not explored here and warrant exploration.”

Reviewer #2

General comment: The way psychosocial stress and depression are discussed is inconsistent throughout the manuscript. In some areas, depression is discussed as a result of stress (one of the research aims of the paper, lines 78-79, 93-94) and in other areas it is considered a measure of stress (lines 71-72, 91-92, 307-308). Similarly, though minor, social support is also occasionally lumped as a ‘psychosocial stress measure’ (e.g., lines 307-308). While this was probably done for ease of communication, it makes it somewhat hard to distinguish whether the references to ‘psychosocial stress’ in the discussion include social support (and also depression).

Response: Thank you for taking the time to review our manuscript. We agree that distinguishing depression as a downstream consequence of stress is key to our conceptual model, and should be referred to as such throughout the manuscript. We have updated our manuscript throughout so that we are consistently differentiating depression from measures of stress. Likewise, we have updated our manuscript to reflect that other stress measures are consistently referred to as such.

Specific Comments:

1. Line 107 – Were women with preeclampsia/gestational hypertension also excluded? Some studies suggest that stress is associated with preeclampsia/gestational diabetes and development of other medical/pregnancy complications, so this may have hindered your ability to detect an association between stress and PTB and could be included as a limitation. Just speculating, but it may also partially explain why your distribution of CES-D scores was lower than what has been reported in other Puerto Rican samples – your sample is inherently healthier than the general population.

Response: Women with preeclampsia and gestational hypertension were not excluded from our study population, although the percent of women with preeclampsia (3%) and gestational hypertension (2.3%) were low. We have added this information to the methods section lines 116-118 with the sentence “Women who developed preeclampsia (3%) and gestational hypertension (2.3%) were not excluded from PROTECT.” 

Additionally, we agree. Because we excluded pregnant women with a variety of complications in pregnancy, we may have unintentionally selected a low-stress cohort. The PROTECT cohort was originally designed to examine environmental risk factors for preterm birth and as such, conditions that were a priori known to increase the risk of preterm birth, such as diabetes and multiple births, were excluded to focus on spontaneous preterm birth, rather than medically-related. We have noted that this was the overall goal of the PROTECT cohort on lines 118-118 with the phrase: “PROTECT was originally designed to examine environmental risk factors for PTB. Therefore, conditions that were a priori known to increase the risk of PTB, such as medical conditions and twinning, were excluded to focus on spontaneous PTB, rather than medically-related.” We have also noted as a limitation that we may have unintentionally selected a low-stress cohort on lines 408-410: “Third, women with preexisting medical complications were excluded from our study population, which may have resulted in our study population being inherently healthier than the general population.”

2. Line 119-120: The first part of this sentence says that the number of events was summed, but then I’m not sure why you would have to take the absolute value of this – it should already be positive, unless you summed the ‘negativity score.’ Clarification of this sentence would be helpful. You could also include the theoretical range to illustrate how the sum was calculated (0-39 for number of events vs. 0 to 117 for number + severity). 

Response: We apologize for the confusion regarding how the Life Experience Survey summary score was calculated. We summed the negative score (all responses coded as -3, -2, or -1) and then took the absolute value of that negative number. We have elaborated on this in lines 130-133 for clarification with the below sentences.

“The number of events that each participant perceived as negative (coded -3, -2, -1) were summed to obtain a negative summary measure, indicative of the perceptions of negative events. The absolute value of the summary measure, indicative of perceptions of negative events, was taken to create a positive, continuous measure of negative life experiences (range 0-28).”

3. Line 165-166: Please indicate whether the Spanish translations of survey instruments have been validated.

Response: The Spanish translations have not been validated, although the Spanish version of the CES-D and PSS have been used in other studies. References which have used Spanish versions of these survey instruments are provided below. We have additionally noted in the limitations section on lines 414-415 that these survey instruments were have not been validated in Spanish: “Lastly, Spanish translations of the LES, NP, PSS, CES-D, and ESSI have not been validated in Puerto Rican populations.”

Glass, T, Leon, CFMD, Bassuk, H and Berkman, LF. 2006. Social engagement and depressive symptoms in late life. Journal of Aging and Health, 18(4): 604–628.

Falcon LM, Todorova I, Tucker K. Social support, life events, and psychological distress among the Puerto Rican 

population in the Boston area of the United States. Aging & mental health. 2009;13(6):863-73.

4. Lines 225-226: It would be helpful to include a study inclusion flowchart.

Response: We have added a study inclusion flow chart as Figure S1. This flow chart is referenced on line 246. 

5. Was your sample different than the general PROTECT cohort? You provide descriptive statistics for those with missing data in table S1, but this is only for those who completed ≥1 measure – what about those who didn’t complete any of the five measures?

Response: Our study population did not differ from the general PROTECT cohort and we have noted this in the results section on lines 250-252. We have added an additional table (Table S1) that shows the distribution of demographics among participants who have complete information on all five measures (N=841). There were 6 participants who were missing information on all psychosocial stress measures. Given the lower percentage of participants missing all stress measures, we did not provide demographic information for this group. 

6. Please report the prevalence of PTB in the sample in Table 1. You could also compare this to a recent estimate for Puerto Rico – may help to indicate whether the inclusion criteria resulted in an overall healthier sample than the general population

Response: We have added the prevalence of PTB to Table 1 (10.2%). We have also noted the 2018 PTB rate for Puerto Rico (11.9%) in the introduction on lines 72-73: “The PTB rate in Puerto Rico declined to 11.9% in 2018, which remains high relative to the mainland U.S.”

7. Table 4: It would be helpful to include a footnote indicating how the categories were defined for all of the continuous stress measures (tertiles per line 215). It would also be helpful to provide the tertile cutoffs to facilitate comparison to other studies.

Response: We have added a footnote to Table 4 indicating that continuous stress measures were categorized into tertiles. The Table 4 footnote also includes the tertile cut points for continuous scales. 

8. Another limitation is potential selection bias introduced by limiting participants to those without pregnancy/medical complications – excluding those who potentially have more psychosocial stress and are at increased risk of preterm birth.

Response: We have acknowledged this as a limitation in the discussion section in lines 408-413 with the sentences: “Third, women with preexisting medical complications were excluded from our study population, which may have resulted in our study population being inherently healthier than the general population. This may have hindered our ability to detect associations between psychosocial stress and PTB, as previous literature has shown that pregnancy complications are associated with elevated stress levels and women with pregnancy complications are at an elevated risk of PTB.”

9. May also want to mention strict confounding assumptions for mediation analysis as a limitation

Response: We have acknowledged the confounding assumptions of mediation analyses as an additional limitation in lines 413-414: “In addition, mediation analysis assumes that there are no unmeasured confounders, which may not be the case here.”

---

## [Editor Report · Decision Letter 1]

6 Jan 2020

Relationships Between Psychosocial Factors During Pregnancy and Preterm Birth in Puerto Rico

PONE-D-19-21339R1

Dear Dr. Ferguson,

We are pleased to inform you that your manuscript has been judged scientifically suitable for publication and will be formally accepted for publication once it complies with all outstanding technical requirements.

With kind regards,

Manisha Nair, DPhil, MSc, MBBS

Academic Editor

PLOS ONE
---

## [Editor Report · Acceptance letter]

9 Jan 2020

PONE-D-19-21339R1 

Relationships Between Psychosocial Factors During Pregnancy and Preterm Birth in Puerto Rico 

Dear Dr. Ferguson:

I am pleased to inform you that your manuscript has been deemed suitable for publication in PLOS ONE. Congratulations! Your manuscript is now with our production department. 

With kind regards,

on behalf of

Dr. Manisha Nair 

Academic Editor

PLOS ONE